# Adaptive output steps: FlexiSteps network for dynamic trajectory prediction

**Yunxiang Liu**<sup></sup>, **Hongkuo Niu**◉⊙*, **Jianlin Zhu**‡

Faculty of Intelligence Technology, Shanghai Institute of Technology, Shanghai, Shanghai, China

⊙ These authors contributed equally to this work.
‡ JZ also contributed equally to this work.

* 236142132@mail.sit.edu.cn

**Data availability statement:** The data underlying this article are available in Argoverse 1 Motion Forcasting Dataset and INTERnational, Adversarial and Cooperative moTION Dataset, at

## Abstract

Accurate trajectory prediction is vital for autonomous driving, robotics, and intelligent decision-making systems, yet traditional models typically rely on fixed-length output predictions, limiting their adaptability to dynamic real-world scenarios. In this paper, we introduce the FlexiSteps Network (FSN), a novel framework that dynamically adjusts prediction output time steps based on varying contextual conditions. Inspired by recent advancements addressing observation length discrepancies and dynamic feature extraction, FSN incorporates a pre-trained Adaptive Prediction Module (APM) to intelligently determine optimal prediction horizons and a Dynamic Decoder (DD) module that enables flexible output generation across different time steps. Additionally, to balance prediction horizon and accuracy, we design a scoring mechanism that leverages Fréchet distance to evaluate geometric similarity between predicted and ground truth trajectories while considering prediction length, enabling principled trade-offs between prediction horizon and accuracy. Our plug-and-play design allows seamless integration with existing trajectory prediction models. Extensive experiments on benchmark datasets including Argoverse and INTERACTION demonstrate that FSN achieves superior prediction accuracy and contextual adaptability compared to traditional fixed-step approaches.

## Introduction

Trajectory prediction plays an essential role in various critical applications such as autonomous driving, robotics, and intelligent decision-making systems. Accurately predicting the future motion of dynamic agents is fundamental to ensuring safety and efficiency in real-world scenarios. Recent advancements in deep learning have significantly improved the precision of trajectory prediction models [1–10]. However, most existing methods are constrained by a fixed-length prediction horizon, limiting their adaptability and effectiveness when confronted with dynamic and unpredictable environments.

Traditional trajectory prediction models typically employ fixed-length output predictions, fundamentally limiting their ability to adapt to dynamic real-world environments. This rigid approach fails to recognize that optimal prediction horizons naturally vary based on changing contextual conditions such as road geometry, traffic density, and agent behaviors.

https://dx.doi.org/10.48550/arXiv.1911.02620
and https://dx.doi.org/10.48550/
arXiv.1910.03088.

**Funding:** The author(s) received no specific
funding for this work.

**Competing interests:** The authors have
declared that no competing interests exist.

Consequently, these models either generate unnecessarily long predictions in simple scenarios or produce insufficient forecasts when longer horizons are needed for complex situations. While recent studies have begun addressing input-side variability through methods like FlexiLength Network (FLN) [11], which introduced calibration and adaptation mechanisms to handle varying observation lengths as shown in Fig 2, the critical issue of output-length adaptability—dynamically determining how far into the future to predict based on current conditions—remains largely unaddressed in existing literature.

Similarly, Length-agnostic Knowledge Distillation (LaKD) [12] was proposed to handle varying observation lengths by dynamically transferring knowledge between trajectories of differing lengths, thus overcoming the inherent limitations of traditional fixed-length input methods. LaKD's approach highlighted that the effectiveness of longer observed trajectories could sometimes be compromised by interference, emphasizing the necessity for adaptive mechanisms to handle real-world trajectory variations.

Inspired by these developments, we propose the FlexiSteps Network (FSN), a novel trajectory prediction framework specifically designed to dynamically adjust the prediction output steps based on contextual cues and environmental conditions. FSN incorporates an innovative pre-trained Adaptive Prediction Module (APM) to intelligently evaluate and determine the optimal number of predicted future steps during inference, ensuring predictions are contextually appropriate. The framework addresses the fundamental limitation of fixed-step predictions by enabling adaptive output generation that responds to varying scenario complexities. Furthermore, to ensure seamless integration with existing architectures, we design a Dynamic Decoder (DD) module that trains multiple specialized decoders for different step lengths, facilitating flexible output generation during inference. This approach significantly enhances prediction adaptability and accuracy across diverse scenarios.

However, the challenge of balancing prediction horizon and accuracy remains a critical issue. Traditional metrics such as Average Displacement Error (ADE) and Final Displacement Error (FDE) often fail to capture the overall shape and temporal consistency of predicted trajectories, leading to suboptimal evaluation of model performance [13]. Recent researches [14,46] and our experiments, as shown in Fig 1, also show that prediction accuracy and prediction horizon are inversely proportional. To address this, we introduce a scoring mechanism that leverages the Fréchet distance, a robust geometric measure that comprehensively assesses trajectory similarity by considering both spatial and temporal relationships [13,15]. By combining the Fréchet distance with the prediction steps, we can effectively evaluate the quality of predictions while dynamically adjusting the output horizon.

This scoring mechanism not only enhances the evaluation of trajectory predictions but also provides a means to trade off between prediction horizon and accuracy. By incorporating the Fréchet distance, we can ensure that the model does not favor shorter-term predictions for higher precision, thus maintaining a balance between flexibility and accuracy.

Through extensive experiments on prominent benchmark datasets, including Argoverse and INTERACTION, our FSN demonstrates significant improvements in flexibility and predictive accuracy compared to traditional models. This framework provides a practical solution to the critical need for adaptive, context-aware trajectory prediction models, setting a new benchmark for effectiveness in dynamic prediction scenarios.

In summary, our work has the following contributions:

- We propose a novel dynamic trajectory prediction framework FlexiSteps Network (FSN), which enables adaptive output step determination based on varying contextual conditions, addressing a critical limitation of traditional fixed-step prediction models. Both

A

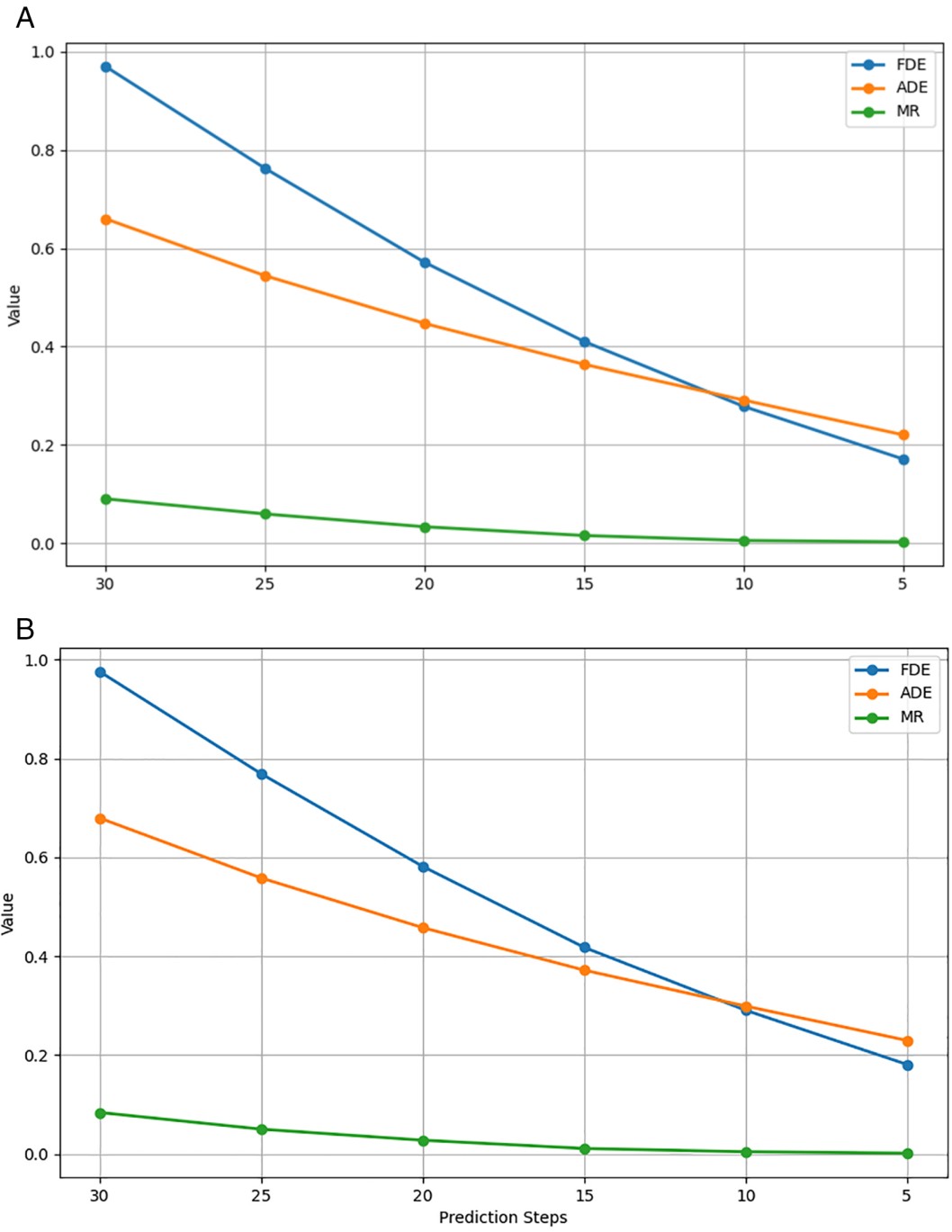

B

**Fig 1. Prediction results of HiVT and HPNet with different fixed prediction steps.**

the Adaptive Prediction Module (APM) and Dynamic Decoder (DD) are designed as plug-and-play components for seamless integration with existing learning-based models.

- We introduce a comprehensive scoring mechanism that enables principled trade-offs between prediction accuracy and temporal horizon, representing the first systematic exploration of metrics to balance prediction step length and accuracy. By incorporating the

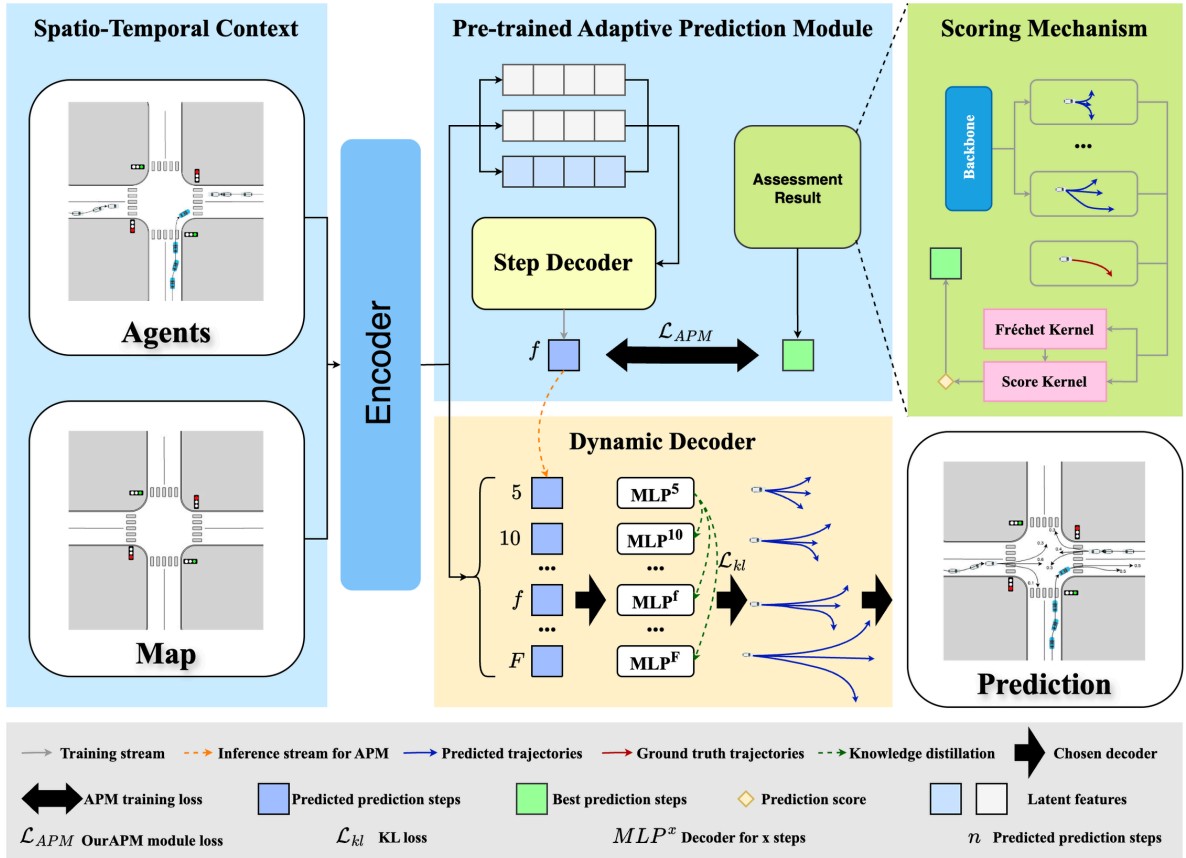

**Fig 2. The overview of our FlexiSteps Network(FSN).** The FSN framework consists of three main components: the pre-trained Adaptive Prediction Module (APM), the Dynamic Decoder (DD), and the scoring mechanism. The APM is responsible for dynamically adjusting the prediction output steps based on contextual cues and environmental conditions. It is trained through the assessment result from trained baseline model, we also take it as the backbone as shown in Scoring Mechanism, in different fixed output steps. Then DD handles varying prediction lengths, allowing the model to output sequences with different step lengths during inference. Finally, the scoring mechanism evaluate the quality of trajectory predictions, considering both spatial and temporal relationships.

Fréchet distance, which considers both spatial and temporal trajectory relationships, our approach provides a robust evaluation framework for prediction quality across varying time horizons, addressing limitations of traditional point-wise distance metrics.

- We validate the accuracy and flexibility through comprehensive experiments on benchmark datasets including Argoverse [16] and INTERACTION [17].

## Related work

### Traditional trajectory prediction

Trajectory prediction plays a critical role in applications such as autonomous driving, robotics, and intelligent systems. Traditionally, trajectory prediction approaches focus on using deep neural networks to learn from historical agent movements and contextual data, effectively modeling complex interactions and behaviors. Methods leveraging Graph Neural Networks (GNNs) [18–22], Generative Adversarial Networks (GANs) [23,24], and Transformer-based architectures have significantly advanced the field [3,25–32], demonstrating robust predictive capabilities on various benchmarks.

## Variable timestep prediction

A major limitation of traditional trajectory prediction methods is their inability to handle varying data effectively. Xu and Fu [11] propose the FlexiLength Network (FLN), which integrates trajectory data of diverse length and employs FlexiLength Calibration (FLC) and Adaptation (FLA) to learn temporal invariant representations. In contrast, Li et al. [12] introduce LaKD, a length-agnostic knowledge distillation framework that dynamically transfers knowledge between trajectories of different lengths. LaKD addresses the limitations of FLN by employing a length-agnostic knowledge distillation to dynamically transfer knowledge and a dynamic soft-masking mechanism to prevent knowledge conflicts. However, both FLN and LaKD primarily focus on input length variability and do not address the challenge of dynamically adjusting output prediction steps, which is critical for real-time decision-making.

Other similar methods such as [33] also try to select different prediction models by understanding different scenarios, but they can not achieve plug-and-play effects for more learning-based methods, which have great limitations. And [7] just divided the prediction into 2 key horizons including short-term and long-term, which is not flexible enough to adapt to the real-world scenarios. In contrast, our proposed FlexiSteps Network (FSN) introduces a pre-trained Adaptive Prediction Module (APM) that dynamically adjusts the output prediction steps based on contextual cues, ensuring optimal prediction accuracy and efficiency. This approach allows for greater flexibility in handling varying prediction requirements, making FSN a more adaptable solution for dynamic trajectory prediction tasks.

## Metrics for trajectory prediction

Traditional metrics such as ADE, FDE, MR, has been used in almost all trajectory prediction methods [19,25,26,34–39] in autonomous driving scenarios. However, these metrics overlook the overall shape and temporal consistency [13]. Song et al. [40] employs Hausdorff Distance [41–43] in their trajectory matching module to ensure coherence. While the Hausdorff distance serves as a natural metric for computing curves or compact sets, it ignores both the direction and motion dynamics along the curves [44,45].

The Fréchet distance, however, accounts for the order of points, making it a superior measure for assessing curve similarity [44]. Thus it is a robust geometric measure, has been proposed as an alternative to ADE and FDE, providing a more comprehensive evaluation of trajectory prediction quality [14,46,47]. By considering both spatial and temporal relationships, the Fréchet distance offers a more nuanced assessment of trajectory predictions, making it a suitable choice for evaluating dynamic prediction models.

## Method

Our FlexiSteps Network (FSN) is designed to dynamically adjust the prediction output steps based on contextual cues and environmental conditions. The overall architecture of FSN is shown in Fig 2. The key components of FSN include a pre-trained Adaptive Prediction Module (APM), a Dynamic Decoder (DD), and a scoring mechanism that incorporates the Fréchet distance to evaluate trajectory predictions. We detail the scoring mechanism in section Scoring mechanism, APM in section Pretrained adaptive prediction module and DD in section Dynamic decoder respectively.

## Problem formulation

Given the target agent $i$'s locations $[l_i^0, .., l_i^{T-1}, l_i^T]$ in the past $T$ time steps, where $l_i^t \in \mathbb{R}^2$ represents the 2D coordinates of the agent, including vehicles, pedestrians and every traffic

participants within the perceived range, at time step $t$, we aim to predict the future locations $[l_i^{T+1}, ..., l_i^{T+F}]$, where $F$ is the number of future steps to be predicted. We present the historical relative trajectory of agent $i$ as $p_i = \left\{ l_i^t - l_i^{t-1} \right\}_{t=1}^{T}$ in historical time steps. Naturally, the target agent will interact with the context including historical locations of surrounding agents and high-definition map(HD map) that represented as $p_{oth} = [p_0, ...p_{N_a}]$ and $p_\xi = \left\{ p_\xi^1 - p_\xi^0 \right\}_{\xi=0}^{N_m}$, where $N_a$ is the number of surrounding agents within a certain radius, $N_m$ is the number of HD map segments in the same radius, and $p_\xi^0$ and $p_\xi^1$ are the start and end points of the HD map segment $\xi$.

## Overview of FlexiSteps network

The overview of our FlexiSteps Network (FSN) is shown in Fig 2. Most methods in trajectory prediction use the framework of encoder-decoder, where the encoder encodes the historical trajectory and context information into a latent representation and the decoder decodes it into future trajectory. Our FSN also follows this way, and we focus on the decoder method, which means that we use baseline encoder as:

$$e_{i,k} = \phi_{enc}([P_i, c_i], [p_\xi, c_\xi]), \tag{1}$$

where $k$ is the mode index in multi-modal prediction framework, $e_{i,k}$ is the latent representation of the target agent $i$ under the $k$ mode, $\phi_{enc}$ is the encoder function from our baseline methods, including [3,6], $P_i$ is concated from $p_i$ and $p_{oth}$, $c_i$ is the agent attribute including agent heading, velocity and agent type, and $c_\xi$ is the HD map lane segments attribute including lane heading, lane turn direction and whether it is an interaction. The output of the encoder is then fed into our APM and DD module to generate prediction steps and future trajectory predictions.

The FSN framework consists of three main components: the pre-trained Adaptive Prediction Module (APM), the Dynamic Decoder (DD), and the scoring mechanism. The APM is responsible for dynamically adjusting the prediction output steps based on contextual cues and environmental conditions. It is trained through the assessment result from trained baseline model in different fixed output steps. And the assessment result is evaluated by the scoring mechanism. The DD is designed to handle varying prediction lengths, allowing the model to output sequences with different step lengths during inference. This flexibility is crucial for adapting to real-world scenarios where the required prediction horizon may vary significantly. Finally, the scoring mechanism incorporates the Fréchet distance to evaluate the quality of trajectory predictions, considering both spatial and temporal relationships. By combining these components, FSN provides a robust and adaptable solution for dynamic trajectory prediction tasks.

## Scoring mechanism

According to the experiments on HiVT [3] and HPNet [6], as shown in Fig 1 the prediction accuracy and prediction horizon are inversely proportional, which means that the longer the prediction horizon, the lower the prediction accuracy. This will cause our Dynamic Decoder to tend to predict shorter durations for lower prediction errors and influence the performance of our Dynamic Prediction. To address this, we introduce a scoring mechanism that incorporates the Fréchet distance to evaluate the quality of trajectory predictions. As shown in Fig 3.

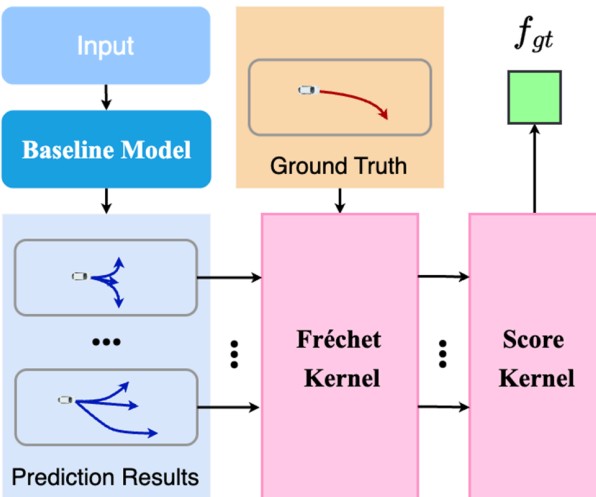

**Fig 3. Overview of our scoring mechanism.** The Fréchet distance is used to evaluate the quality of trajectory predictions, and the prediction length is also considered to balance accuracy and prediction horizon.

**Fréchet distance Kernel.** The Fréchet distance is a robust geometric measure that comprehensively assesses trajectory similarity by considering both spatial and temporal relationships [13,15]. However, it can not be directly applied into machine learning frameworks due to its non-smoothness function as:

$$d_F(X, Y) := \min_{A \in \mathcal{A}} \max_{(i,j) \in A} d(x_i, y_j) \tag{2}$$

where $d(x,y)$ is the Euclidean distance between vector $x$ and $y$, $X = \{x_1, x_2, ..., x_m\}$ and $Y = \{y_1, y_2, ..., y_n\}$ are two sets of points, and $\mathcal{A}$ is the set of all possible alignments between the two sets. Fréchet distance kernal (FRK) [48] design soft-min approximation and smooth-min approximation to make fréchet distance computable in machine learning frameworks. Howereve, the smooth-min approximation of the original FRK is noise-sensitive because the exponential weight assigned to outliers may be too large. To address this issue, we propose a new Fréchet distance kernal (FDK) to approximate the fréchet distance as a smooth function by introducing Huber Loss Smooth [49,50]:

$$H(z, \delta) = \begin{cases} \frac{1}{2}z^2 & \text{if } |z| \leq \delta \\ \delta(|z| - \frac{1}{2}\delta) & \text{otherwise} \end{cases} \tag{3}$$

where $\delta$ is threshold parameters, which is set to 0.1 in our experiments. The Fréchet distance kernal (FDK) is defined as:

$$\min_{(i,j) \in A} \phi(x_i, y_j) \approx \frac{\sum\limits_{(i,j \in A)} \phi(x_i, y_j) \cdot exp(-\beta \cdot H(\phi(x_i, y_j), \delta))}{Z_A} \tag{4}$$

$$FDK(X, Y) := \sum_{A \in \mathcal{A}} \sum_{(i,j) \ inA} \frac{\widetilde{\phi}_\varepsilon(x_i, y_j) exp(-\beta \cdot H(\phi(x_i, y_j), \delta))}{Z_A} \tag{5}$$

where $\phi(x_i, y_j) = exp(-d(x_i, y_j)/\gamma)$, $\widetilde{\phi}_\epsilon(x_i, y_j) := exp(-d(x_i, y_j)/\gamma + \epsilon\delta(x_i - y_j))$ with parameters $\epsilon > 0$, $\beta$, $Z_A > 0$.

**Score Kernel.** In addition, in order to balance accuracy and prediction length, it is not enough to just calculate the fréchet distance, the prediction length is necessary to be included. The smaller the distance, the closer between the prediction trajectories and ground truth trajectories. To make the fraction of the prediction trajectory smaller, the higher the prediction quality, we supplement the prediction length at the denominator as:

$$q_i^f = \frac{d_i^f}{f} \tag{6}$$

where $q_i^f$ is the score of agent $i$ with prediction steps $f$, $d_i^f = FDK(\mu_i^f, gt_i^f)$ is the Fréchet distance between the predicted trajectory $\mu_i^f$ and the ground truth trajectory $gt_i^f$ of agent $i$ with prediction steps $f$. The score $q_i^f$ is designed to be lower for better predictions, as it combines the Fréchet distance with the prediction length, thus penalizing longer predictions that do not match the ground truth well.

Finally, we can get the final score from all prediction steps:

$$q_i = min(q_i^f|_{f=5}^F) \tag{7}$$

And the prediction step of agent $i$ is:

$$f_{gt_i} = \text{argmin}_{f\in\{5,...,F\}} q_i^f \tag{8}$$

To summarize, we express the scoring mechanism in:

$$f_{gt_i} = \phi_{score}(\mu_{i,k}^f, gt_i^f) \tag{9}$$

where $f_{gt_i}$ is the optimal prediction step achieves the lowest score, i.e.the best result, under our scoring mechanism, $\phi_{score}$ is the scoring mechanism function, and $gt_i^f = \{l_i^t\}_{t=T+1}^{T+f}$ is the ground truth future trajectory of agent $i$. To calibrate the trade-off between prediction horizon and trajectory accuracy, we do not introduce additional hyperparameters; instead, the division by the step length $f$ naturally balances longer and shorter predictions. This design ensures that longer horizons are only favored when their trajectory similarity (measured by Fréchet distance) is sufficiently high. We confirmed the stability of this trade-off through empirical validation: the formulation consistently produced reasonable horizon selections across the validation set without requiring further tuning. By combining the Fréchet distance with the prediction steps, we can effectively evaluate the quality of predictions while dynamically adjusting the output horizon.

## Pretrained adaptive prediction module

The Adaptive Prediction Module (APM) is a pre-trained module that adaptively adjusts the prediction steps based on the contextual information and environmental conditions. The APM is trained using a set of historical trajectories and their corresponding future trajectories, allowing it to learn the optimal prediction steps for different scenarios. The training process involves evaluating the performance of the baseline model with different fixed

prediction steps and using the assessment results to guide the APM's adjustments. And the detailed evaluation criteria will illustrated in section Scoring mechanism.

**Training stage.** To train our APM, we first collect the prediction results from the baseline model trained by different fixed output steps:

$$\mu_{i,k}^{f}, b_{i,k}^{f} = \phi_{D^f}(\phi_{enc}([P_i, ci], [p_\xi, c_\xi])) \tag{10}$$

where $\phi_{D^f}$ is a twolayerMLP as the decoder function from our baseline methods, $f \in [1, F]$ is the fixed output steps, $\mu_{i,k}^{f}$ and $b_{i,k}^{f}$ are the predicted future trajectory and probability of agent $i$ mode $k$ with fixed output steps $f$ respectively.

Then we use the scoring mechanism to evaluate the prediction results and return the optimal prediction steps for each agent $f_{gt_i}$. After collecting the optimal prediction steps for all agents among datasets, we can train our APM as illustrated in Fig 4 and following:

$$b_i' = \phi_{step}(e_{i,k}) \tag{11}$$

$$f_i' = \text{argmax}_{j \in \{1,2,\dots,F\}} b_i'^j \tag{12}$$

$$\mathcal{L}_{cls} = -\frac{1}{N} \sum_{i=1}^{N} \sum_{j=1}^{F} b_i^j \log(b_i'^j) \tag{13}$$

$$\mathcal{L}_{reg} = \frac{1}{N} \sum_{i=1}^{N} \left\| f_i' - f_{gt_i} \right\|_2^2 \tag{14}$$

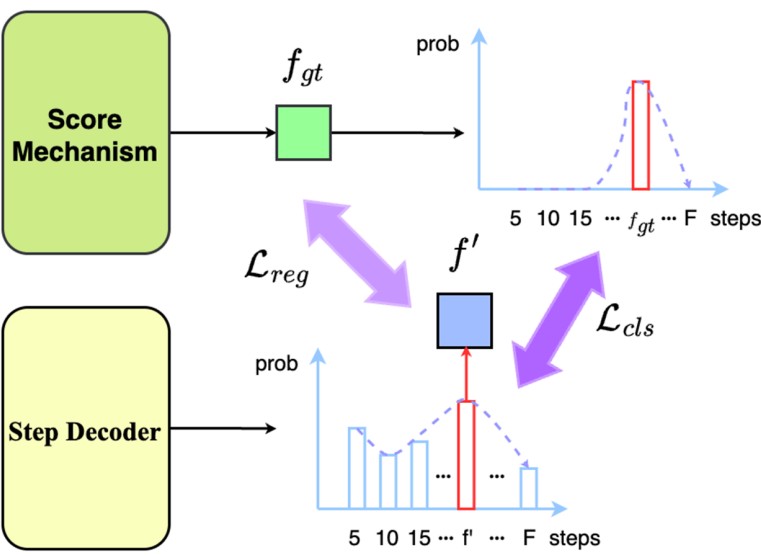

**Fig 4. Training stage of the Adaptive Prediction Module (APM).** The APM is trained to predict the optimal prediction step based on the encoded latent features of the target agent and its context.

where $b_i' = \left\{ b_i'^{,f} \right\}_{f=1}^{F}$ is the predicted probability distribution of different prediction steps for agent $i$, $N$ is the number of agents in the dataset, $\phi_{step}$ is a two-layer-Multilayer-Perceptron(twolayerMLP) as the steps decoder, $\mathcal{L}_{cls}$ is the cross entropy loss function, $\mathcal{L}_{reg}$ is the regression loss, and $b_i$ is the one-hot encoded ground truth with $b_i^j = \mathbb{1}[j = f_i]$. The APM is trained to predict the optimal prediction step based on the input context.

In summary, the loss function for trainging the APM is:

$$\mathcal{L}_{APM} = \mathcal{L}_{cls} + \mathcal{L}_{reg} \tag{15}$$

**Inference stage.** APM is applied to the training and inference process of FSN as a plug-and-play module. APM takes the encoded latent features of the target agent and its context as input and outputs the predicted optimal prediction step:

$$f_i = \phi_{step}(e_{i,k}) \tag{16}$$

where $f_i$ is the predicted optimal prediction step for agent $i$. The APM can be integrated into any trajectory prediction model, allowing it to dynamically adjust the prediction steps based on the contextual information and environmental conditions. This adaptability is crucial for ensuring that the model can effectively handle varying prediction requirements in real-world scenarios.

## Dynamic decoder

Traditional trajectory prediction models typically use MLP or twolayerMLP as the decoder to decode the latent embedding into future trajectory with fixed steps, as shown in Eq 10. This leads to a limitation in flexibility, as the model can only align weights to a specific output steps. To address this, we propose Dynamic Decoder (DD) that can handle varying prediction lengths, allowing the model to output sequences with different step lengths during inference.

Specifically, each time step of prediction matches the decoder with the corresponding step length. During training, the input $e_i$ and its output steps $f_i$, which is from the APM, are processed by their respective sub-networks $\{\phi_{D^f}\}_{f=5}^{F}$ and each $\phi_{D^f}$ parameter update is independent of each other:

$$\mu_{i,k}^{f_i}, b_{i,k}^{f_i} = \phi_{DD}(e_{i,k}, f_i) \tag{17}$$

where $\mu_{i,k}^{f_i}$ and $b_{i,k}^{f_i}$ are the predicted future trajectory and probability of agent $i$ mode $k$ with output steps $f_i$, $\phi_{DD} = \{\phi_{D^f}\}_{f=5}^{F}$ is the decoder function for the whole network. During inference, the sub-network matching the predicted output steps $f_i'$ is exclusively activated.

Additionally, the input $e_{i,k}$ with different movement information may lead to different fit, which has potential drawbacks. To address this, we employ KL divergence [12] to distill knowledge from the "lower" score sequences to the "higher" ones:

$$\mathcal{L}_{KL} = KL(V_h, V_l) \tag{18}$$

where $V_h$ and $V_l$ are the latent features of the "higher" and "lower" score sequences respectively.

## Training objective

In addition to the pre-trained APM, we also need to train the entire FlexiSteps Network. Following [3] and HPNet [6], we also adopt the negative log-likelihood in HiVT and Huber in HPNet as the regression loss $\mathcal{L}_{reg}$. We also use the cross-entropy loss as the classification loss $\mathcal{L}_{cls}$ to optimize the model. Finally, the total loss function can be expressed as:

$$\mathcal{L} = \mathcal{L}_{reg} + \mathcal{L}_{cls} + \lambda\mathcal{L}_{KL} \tag{19}$$

where $\lambda$ is a hyperparameter to balance the contribution of KL loss function. The training process involves optimizing the model parameters to minimize the total loss, allowing the FlexiSteps Network to effectively learn from the data and adapt to varying prediction requirements.

## Experiments

### Settings

**Baselines.** Our FlexiSteps Network (FSN) is a plug-and-play model that can be integrated into deep learning-based trajectory prediction models. We have chosen HiVT [3] and HPNet [6], a typical method and a state-of-the-art open source method, as our baseline models. Due to limited computing resources,for the HPNet model, we only use the propose stage among the propose prediction and refine prediction stage and marked as HPNet(h) below. For more comprehensive evaluation, we include 3 additional baselines: (1) **Isolated Training** (IT): Train the baseline model with different fixed output steps. (2) **Intercepted Result** (IR): The model just trained once with the longest prediction steps and the prediction results are intercepted to different steps. (3) **Fixed Steps Network** (fixed): A version of our FSN where the output of APM is fixed to 5-30 steps during training and inference respectively. This allows us to isolate the impact of dynamic step prediction from other factors.

**Datasets.** **Argoverse** [16] is a large-scale dataset for autonomous driving, containing diverse scenarios and complex interactions between agents. It includes high-definition maps and rich contextual information, making it suitable for evaluating trajectory prediction models. The dataset is divided into training, validation, and test sets, with a total of 324,557 interesting vehicle trajectories. This rich dataset includes high-definition (HD) maps and recordings of sensor data, referred to as "log segments," collected in two U.S. cities: Miami and Pittsburgh. These cities were chosen for their distinct urban driving challenges, including unique road geometries, local driving habits, and a variety of traffic conditions.

**INTERACTION** [17] is an extensive dataset developed for autonomous driving research, particularly focusing on behavioral aspects such as motion prediction, behavior analysis, and behavior cloning. It contains a diverse collection of natural movement patterns from various road users, including vehicles and pedestrians, captured across numerous highly interactive traffic scenarios from multiple countries. This dataset provides valuable information for studying complex agent interactions in different driving environments.

**Metrics.** Although we have metioned the limitations of mainstream metrics such as ADE and FDE, we still use them because we need to compare our FSN with traditional methods to verify the performance of our method.

Specifically, for Argoverse dataset, we use minimum ADE (minADE), minimum FDE (minFDE) and Miss Rate (MR) to measure the accuracy and robustness of the model. The minimum ADE is the minimum average displacement error between the predicted trajectory and the ground truth trajectory, while the minimum FDE is the minimum final displacement

error. The Miss Rate is the percentage of predicted trajectories that do not match any ground truth trajectory within a certain threshold.

For INTERACTION dataset, we employ minJointADE, minJointFDE and Cross Collision Rate (CCR) to evaluate the performance of joint trajectory prediction. The minJointADE is the minimum average displacement error between the predicted joint trajectory and the ground truth joint trajectory, while the minJointFDE is the minimum final displacement error. The Cross Collision Rate (CCR) is the percentage of predicted joint trajectories that collide with each other, indicating the model's ability to handle interactions between agents.

**Implementation details.** For the HiVT baseline, we train our APM and FSN for 64 epochs with a batch size of 32 on 1 NVIDIA A6000 GPU, using the AdamW optimizer with a learning rate of $5 \times 10^{-4}$ and weight decay of $1 \times 10^{-4}$. For the HPNet(h) baseline, we train our APM and FSN for 64 epochs with a batch size of 4 on 1 NVIDIA A6000 GPU, using the AdamW optimizer with a learning rate of $5 \times 10^{-4}$ and weight decay of $1 \times 10^{-4}$.

The hyperparameter $\lambda$ is set to 0.5 for both baselines. The training process involves optimizing the model parameters to minimize the total loss, allowing the FlexiSteps Network to effectively learn from the data and adapt to varying prediction requirements.

## Main results

**Argoverse.** As shown in Table 1, our FlexiSteps Network (FSN) outperforms the baselines on the Argoverse dataset across all prediction steps. Specifically, FSN achieves a minimum ADE of 0.2121 and a minimum FDE of 0.1632 at 5 timesteps, which is significantly better than the IT and IR baselines. The results demonstrate that FSN effectively leverages the pre-trained Adaptive Prediction Module (APM) and Dynamic Decoder (DD) to adaptively adjust the prediction steps based on contextual information, leading to improved accuracy and robustness in trajectory prediction.

Furthermore, the FSN(fixed) variant, which uses fixed prediction steps during training and inference, also outperforms the baseline methods but still falls short of our full FSN approach, highlighting the importance of adaptive prediction step selection in improving model performance (Figs 5 and 6).

In addition, we also compare our FSN with the state-of-the-art dynamic prediction methods FLN and LaKD. The results show that FSN achieves better performance than FLN and LaKD, demonstrating the effectiveness of our approach in handling varying prediction lengths. The minimum ADE and FDE of FSN at 30 timesteps are 0.6571 and 0.9602, respectively, which are significantly lower than those of FLN and LaKD. This indicates that FSN can effectively adapt to different prediction requirements and achieve better performance in trajectory prediction tasks.

**Interation.** As shown in Table 1, our FlexiSteps Network (FSN) also outperforms the baselines on the INTERACTION dataset across all prediction steps. Specifically, FSN achieves a minimum Joint ADE of 0.0066 and a minimum Joint FDE of 0.0067 at 5 timesteps, which is significantly better than the IT and IR baselines. Furthermore, the FSN(fixed) variant also shows improvements over the baseline methods, achieving better performance than IT and IR across different prediction horizons, but still falls short compared to our full adaptive FSN approach. This further demonstrates that FSN effectively leverages the pre-trained Adaptive Prediction Module (APM) and Dynamic Decoder (DD) to adaptively adjust the prediction steps based on contextual information, leading to improved accuracy and robustness in joint trajectory prediction.

**Table 1. Performance comparison of FlexiSteps Network (FSN).**

| Dataset | Method | Metrics FDE/ADE/MR | | | | | |
|---|---|---|---|---|---|---|---|
| | | 5 timesteps | 10 timesteps | 15 timesteps | 20 timesteps | 25 timesteps | 30 timesteps |
| Argoverse 1 | HiVT-IT [3] | 0.1710/0.2203/0.0024 | 0.2782/0.2910/0.0052 | 0.4103/0.3638/0.0153 | 0.5714/0.4470/0.0331 | 0.7626/0.5442/0.0592 | 0.9701/0.6602/0.0923 |
| | HiVT-IR [3] | 0.3635/0.2904/0.0107 | 0.5244/0.3753/0.0240 | 0.6701/0.4542/0.0433 | 0.8038/0.5283/0.0672 | 0.9219/0.5974/0.0915 | 0.9698/0.6611/0.0921 |
| | HiVT-FLN [11] | - | - | - | - | - | 1.0325/0.7026/0.1033 |
| | HiVT-LaKD [12] | - | - | - | - | - | 0.9864/0.6807/0.0928 |
| | HiVT-FSN(fixed) | 0.1698/0.2187/0.0022 | 0.2751/0.2883/0.0050 | 0.4052/0.3589/0.0148 | 0.5651/0.4412/0.0325 | 0.7553/0.5447/0.0583 | 0.9654/0.6621/0.0932 |
| | HiVT-FSN(ours) | **0.1632/0.2121/0.0020** | **0.2678/0.2804/0.0045** | **0.3952/0.3501/0.0129** | **0.5503/0.4302/0.0287** | **0.7501/0.5393/0.0512** | **0.9602/0.6571/0.0904** |
| Argoverse 1 | HPNet(h)-IT [6] | 0.1621/0.2102/0.0023 | 0.2990/0.2905/0.0045 | 0.4177/0.3717/**0.0111** | 0.5813/0.4576/0.0278 | 0.7685/0.5579/0.0501 | 0.9751/0.6790/0.0838 |
| | HPNet(h)-IR [6] | 0.3818/0.3037/0.0109 | 0.5491/0.3933/0.0238 | 0.6924/0.4747/0.0419 | 0.8232/0.5490/0.0657 | 0.9340/0.6169/0.0873 | 0.9741/0.6784/0.0829 |
| | HPNet(h)-FSN(fixed) | 0.1627/0.2098/0.0021 | 0.2964/0.2887/0.0046 | 0.4145/0.3692/0.0114 | 0.5778/0.4551/0.0277 | 0.7632/0.5543/0.0493 | 0.9705/0.6723/0.0801 |
| | HPNet(h)-FSN(ours) | **0.1583/0.2064/0.0019** | **0.2853/0.2862/0.0042** | **0.4001/0.3602/0.0118** | **0.5726/0.4403/0.0269** | **0.7574/0.5504/0.0487** | **0.9624/0.6678/0.0786** |
| INTERACTION | HiVT-IT [3] | 0.0067/0.0069 | 0.0197/0.0153 | 0.0587/0.0245 | 0.1577/**0.0578** | 0.3058/0.1104 | 0.5963/0.2034 |
| | HiVT-IR [3] | 0.0140/0.0087 | 0.0368/0.0179 | 0.0930/0.0347 | 0.1980/0.0644 | 0.3577/0.1096 | 0.5954/0.2057 |
| | HiVT-FSN(fixed) | 0.0068/0.0070 | 0.0185/0.0145 | 0.0523/0.0214 | 0.1571/0.0579 | 0.2901/0.1127 | 0.5932/0.1993 |
| | HiVT-FSN(ours) | **0.0067/0.0066** | **0.0172/0.0139** | **0.0496/0.0186** | **0.1496/0.0584** | **0.2874/0.1058** | **0.5829/0.1927** |
| INTERACTION | HPNet(h)-IT [6] | 0.0068/0.0069 | 0.0205/0.0146 | 0.0634/0.0280 | 0.1501/0.0549 | 0.3008/0.1053 | 0.5721/0.1761 |
| | HPNet(h)-IR [6] | 0.0148/0.0092 | 0.0388/0.0188 | 0.0962/0.0364 | 0.2029/0.0670 | 0.3625/0.1133 | 0.5754/0.1765 |
| | HPNet(h)-FSN(fixed) | 0.0069/0.0068 | 0.0198/0.0131 | 0.0563/0.0225 | 0.1402/0.0493 | 0.2989/0.1005 | 0.5683/0.1721 |
| | HPNet(h)-FSN(ours) | **0.0064/0.0066** | **0.0194/0.0122** | **0.0542/0.0201** | **0.1378/0.0468** | **0.2976/0.0936** | **0.5645/0.1647** |

Performance comparison of FlexiSteps Network (FSN) with baselines on Argoverse and INTERACTION datasets. The best results are highlighted in bold. The "-" indicates that the method does not open source and the results are not available.

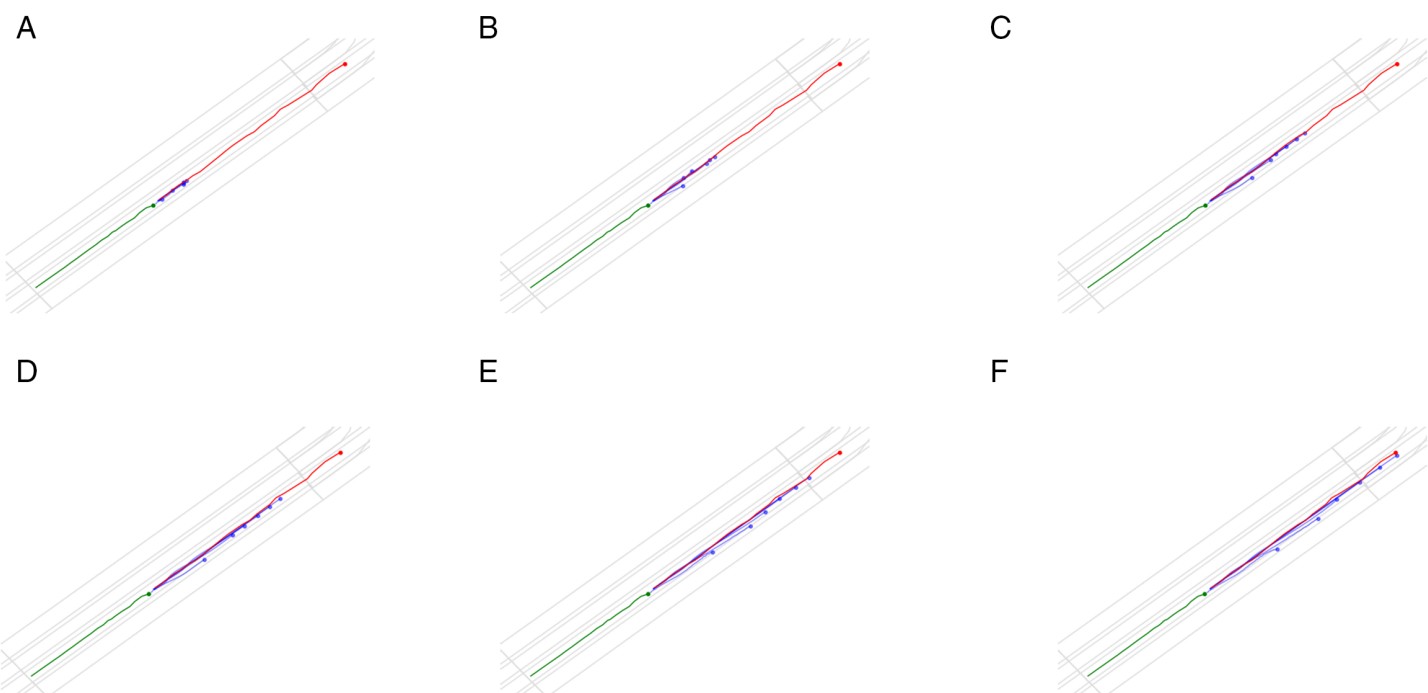

**Fig 5. The prediction results of HiVT with different prediction steps from Intercepted Results (IR).** The blue line is the prediction trajectories, the red line is the ground truth trajectory, and the green line is the historical trajectory.

**Computational efficiency analysis.** As shown in Table 2, we evaluate the computational efficiency of our FlexiSteps Network (FSN) by measuring the inference time and the number of parameters when integrated with HiVT and HPNet(h) as backbone models. The results indicate that FSN introduces a slight increase in inference time and parameters compared to the original backbone models. Specifically, when using HiVT as the backbone, FSN increases the average inference time from 42.56 ms to 45.58 ms and the number of parameters from 2.5M to 3.1M. Similarly, with HPNet(h) as the backbone, FSN increases the average inference time from 61.72 ms to 64.41 ms and the number of parameters from 4.1M to 4.7M.

The results in Table 2 reveal that FSN introduces a moderate increase in computational resources compared to baseline models. For inference time, FSN adds approximately 3.02 ms to HiVT and 2.69 ms to HPNet(h). This increase can be attributed to the additional processing required by the Adaptive Prediction Module (APM), which must dynamically evaluate the appropriate prediction horizon during inference. More significantly, FSN increases the parameter count by 0.6M for both backbones. This substantial parameter increase primarily stems from the Dynamic Decoder (DD) design, which maintains multiple specialized fixed-length decoders to handle different prediction horizons. While each individual decoder has relatively modest parameter requirements, maintaining a comprehensive set of decoders for various prediction lengths (from 5 to 30 timesteps) collectively contributes to the significant parameter increase. Despite these costs, the performance improvements demonstrated in our experiments suggest that this computational trade-off is justified for applications requiring adaptive prediction horizons.

**Distribution of prediction steps.** demonstrate the adaptability of our FlexiSteps Network (FSN) in predicting varying output steps, we analyze the distribution of predicted steps

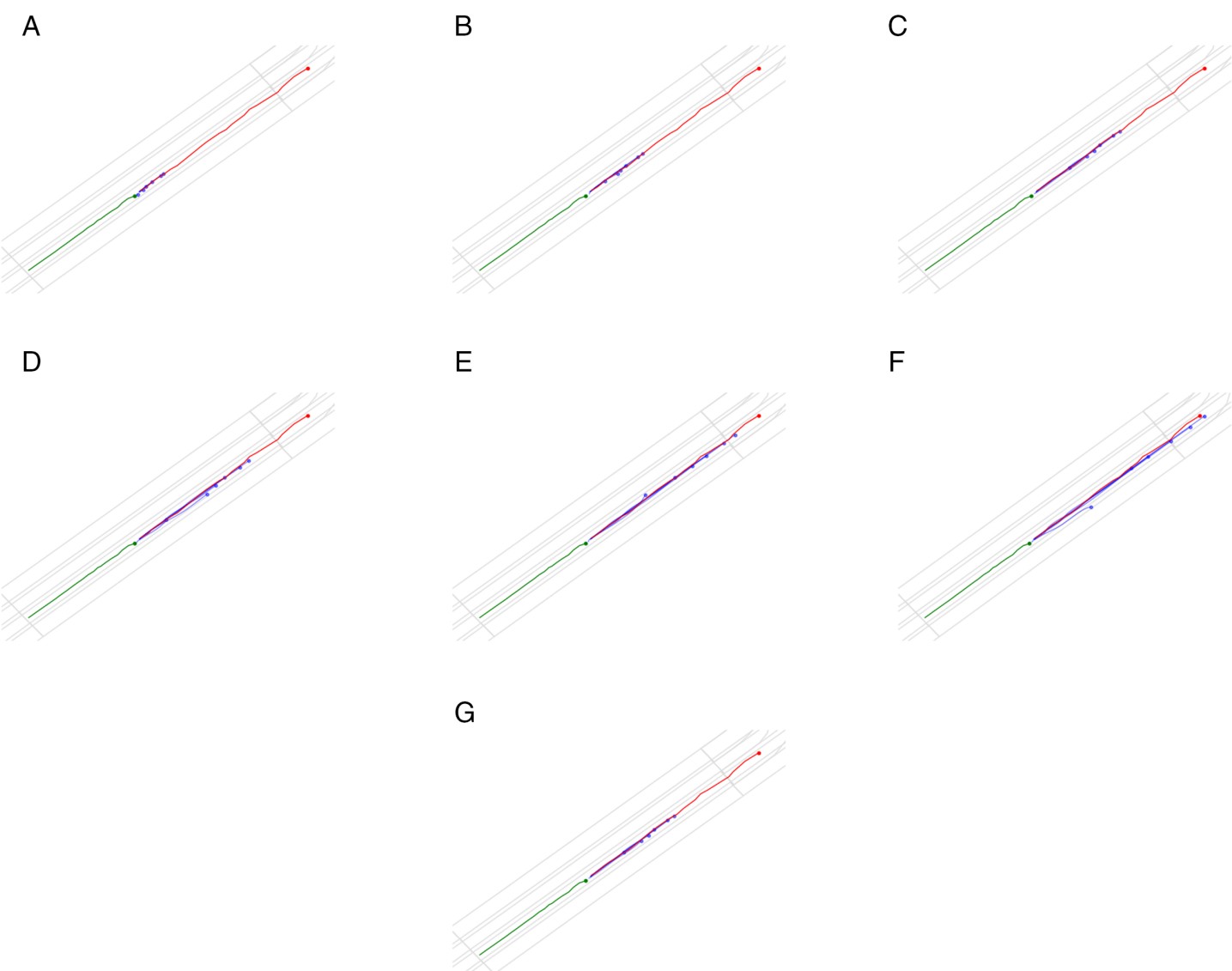

**Fig 6. The prediction results of HiVT trained with fixed prediction steps.** The blue line is the prediction trajectories, the red line is the ground truth trajectory, and the green line is the historical trajectory. The last subfigure is the prediction results of our FlexiSteps Network (FSN).

**Table 2. Computational Efficiency Analysis of FlexiSteps Network (FSN).**

| Model | Inference time(ms) | | | Parameters |
|---|---|---|---|---|
| | avg | min | max | |
| HiVT | 42.56 | 27.50 | 269.76 | 2.5M |
| HiVT-FSN | 45.58 | 27.27 | 306.74 | 3.1M |
| HPNet(h) | 61.72 | 32.23 | 453.87 | 4.1M |
| HPNet(h)-FSN | 64.41 | 34.06 | 402.09 | 4.7M |

We test the inference time and parameters of our FlexiSteps Network (FSN) with HiVT and HPNet(h) as our back-bone on Argoverse validation dataset. "avg", "min", and "max" represent the average, minimum, and maximum inference time in milliseconds, respectively.

on the Argoverse validation dataset. As shown in Fig 7, in small-scale scenarios with fewer than 10 agents, longer horizons (25-30 steps) remain prevalent, reflecting relatively simple interaction dynamics. For medium-scale scenarios (11-30 agents), the majority of cases concentrate on 15-20 steps, while still preserving both shorter and longer horizons, indicating

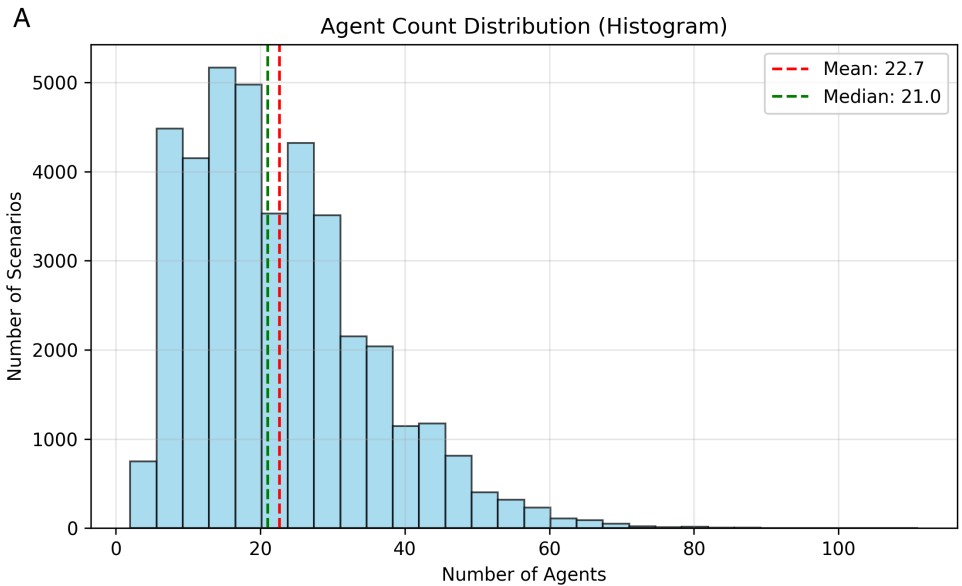

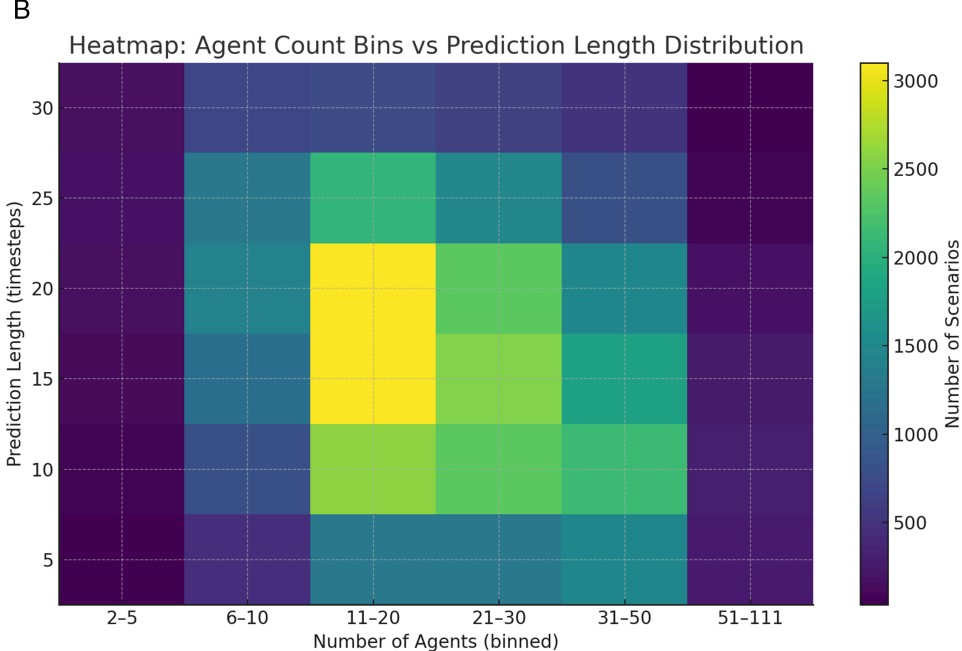

**Fig 7. Prediction Steps Distribution on Argoverse validation set.** (a) is the distribution of the number of agents in different scenarios of Argoverse validation. (b) is a heatmap showing the distribution of prediction lengths across different agent count bins. The x-axis represents the number of agents, while the y-axis represents the prediction length in the scenario. The color intensity indicates the number of scenarios for each combination of agent count and prediction length.

diverse prediction demands. As the number of agents increases beyond 30, shorter horizons (5-15 steps) dominate, and in highly congested scenes (51+ agents), predictions are largely restricted to very short lengths. Overall, the distribution highlights that medium horizons (10-20 steps) account for the majority of scenarios, aligning with the intuition that moderate predictive ranges balance stability and adaptability in complex environments.

## Ablation study

To evaluate the effectiveness of each component in our FlexiSteps Network (FSN), we conduct an ablation study on the Argoverse validation dataset with 30 prediction steps. The results are presented in Table 3. We analyze the impact of the Dynamic Decoder (DD), and scoring mechanism on the overall performance of FSN. Specifically, we compare the performance of Dynamic Decoder (DD) with and without the KL divergence loss, and the performance of the scoring mechanism with and without the Fréchet distance, we employ ADE and FDE instead of Fréchet distance.

The results show that our score mechanism with Fréchet distance achieves the best performance, with a minimum FDE of 0.9602 and a minimum ADE of 0.6571, significantly outperforming the score mechanism with ADE and FDE. This indicates that the Fréchet distance effectively captures the spatial and temporal relationships in trajectory predictions, leading to improved accuracy. And the Dynamic Decoder (DD) with KL divergence loss also improves the performance, achieving a minimum FDE of 0.9683 and a minimum ADE of 0.6647, compared to the DD without KL divergence loss. This demonstrates that the KL divergence loss helps to distill knowledge from lower score sequences to higher ones, enhancing the model's ability to adapt to varying prediction lengths.

## Conclusion

In this paper, we propose the FlexiSteps Network (FSN), a novel framework for dynamic trajectory prediction that adapts the prediction output steps based on contextual cues and environmental conditions. FSN consists of three main components: a pre-trained Adaptive Prediction Module (APM), a Dynamic Decoder (DD) module, and a scoring mechanism that incorporates the Fréchet distance to evaluate trajectory predictions. The APM is trained to predict the optimal prediction step based on the encoded latent features of the target agent and its context, while the DD handles varying prediction lengths. The scoring mechanism evaluates the quality of trajectory prediction by combining the Fréchet distance with the prediction length. Our experimental results demonstrate that FSN outperforms state-of-the-art methods on both Argoverse and INTERACTION datasets, achieving better accuracy trajectory prediction tasks.

**Table 3. Ablation study of FlexiSteps Network (FSN).**

| Score | | | Dynamic Decoder | | FDE | ADE | MR |
|---|---|---|---|---|---|---|---|
| Fréchet | FDE | ADE | w/o KL | w/ KL | | | |
| | ✓ | | | | 0.9705 | 0.6602 | 0.0922 |
| | | ✓ | | | 0.9727 | 0.6594 | 0.0923 |
| ✓ | | | ✓ | | 0.9683 | 0.6647 | 0.0921 |
| ✓ | | | | ✓ | **0.9602** | **0.6571** | **0.0904** |

Ablation study of FlexiSteps Network (FSN) with HiVT as our backbone on Argoverse validation dataset. The results show the impact of different components on the overall performance of FSN. The best results are highlighted in bold.

While our FlexiSteps Network (FSN) shows promising results in dynamic trajectory prediction, there are still several limitations and potential areas for future work. One limitation is the reliance on the pre-trained Adaptive Prediction Module (APM), which may not generalize well to unseen scenarios or environments. Future work could explore methods to enhance the adaptability of APM, such as incorporating online learning or transfer learning techniques. Additionally, the scoring mechanism based on Fréchet distance may not capture all aspects of trajectory quality, and further research could investigate alternative metrics or hybrid approaches to improve evaluation accuracy. Finally, while FSN demonstrates flexibility in handling varying prediction lengths, it may still struggle with highly dynamic or unpredictable environments. Future research could focus on enhancing the model's robustness to such scenarios, potentially through the integration of reinforcement learning or other adaptive techniques.

## Author contributions

**Conceptualization:** Yunxiang Liu, Hongkuo Niu.

**Data curation:** Hongkuo Niu.

**Formal analysis:** Hongkuo Niu.

**Investigation:** Hongkuo Niu.

**Methodology:** Hongkuo Niu.

**Project administration:** Hongkuo Niu.

**Resources:** Hongkuo Niu.

**Software:** Hongkuo Niu.

**Supervision:** Yunxiang Liu, Jianlin Zhu.

**Validation:** Hongkuo Niu.

**Visualization:** Hongkuo Niu.

**Writing – original draft:** Hongkuo Niu.

**Writing – review & editing:** Hongkuo Niu.

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
