## [Decision Letter · Decision Letter 0]

4 Sep 2025

PONE-D-25-40072Adaptive Output Steps: FlexiSteps Network for Dynamic Trajectory PredictionPLOS ONE

Dear Dr. Niu,

Thank you for submitting your manuscript to PLOS ONE. After careful consideration, we feel that it has merit but does not fully meet PLOS ONE’s publication criteria as it currently stands. Therefore, we invite you to submit a revised version of the manuscript that addresses the points raised during the review process.

We look forward to receiving your revised manuscript.

Kind regards,

Jinhao Liang

Academic Editor

PLOS ONE

Additional Editor Comments:

Please revise the manuscript in a point-by-point manner, with particular emphasis on addressing the reviewers’ concerns regarding novelty from prior work.

Reviewers' comments:

Reviewer's Responses to Questions

**Comments to the Author**

1. Is the manuscript technically sound, and do the data support the conclusions?

Reviewer #1: Partly

Reviewer #2: Partly

2. Has the statistical analysis been performed appropriately and rigorously? 

Reviewer #1: N/A

Reviewer #2: N/A

3. Have the authors made all data underlying the findings in their manuscript fully available?

Reviewer #1: No

Reviewer #2: Yes

4. Is the manuscript presented in an intelligible fashion and written in standard English?

Reviewer #1: No

Reviewer #2: Yes

5. Review Comments to the Author

Reviewer #1: 1. What is the research subject of this article - vehicle trajectory prediction or pedestrian trajectory prediction? The author needs to be clear.

2. The innovation of this article is not clear. In the contribution section, the author fails to distinguish the differences between the methods presented in this article and those of previous studies, nor does he highlight the problems that this method can solve.

3. The font in Fig.2 is not uniform. The picture needs to be adjusted. The color of the arrows needs to be darkened. The meanings of some variables in the picture are unclear. Besides, what is the input of the scoring mechanism? It cannot be seen from the picture. Overall, Fig.2 is confusing.

4. The full names of APM and DD should be provided for the first time. Additionally, I haven't found the structural details of APM and DD, which prevents me from evaluating the innovativeness of these two modules.

5. The experimental part is insufficient and the comparison methods used are limited. The author claims that the method in this paper improves efficiency, but no analysis of computational cost has been presented.

6. Many typos exist in this work. Please double-check it.

Reviewer #2: The authors propose FlexiSteps Network (FSN), a novel framework for trajectory prediction that dynamically adjusts the number of future time steps predicted based on contextual conditions. This addresses a critical limitation of fixed-horizon models and has strong potential for real-world deployment in autonomous systems.

The motivation is excellent, the use of Fréchet distance for geometric-temporal evaluation is well-justified, and experiments on Argoverse and INTERACTION add credibility. However, several revisions are needed to strengthen the technical contribution and ensure reproducibility:

Clarify the Adaptive Prediction Module (APM):

What is its architecture (e.g., MLP, Transformer)?

How is it trained? Are labels for optimal step length generated from ground truth trajectories or heuristics?

Is it pre-trained independently or jointly with the decoder?

Define the scoring mechanism explicitly:

Provide the mathematical formula for the score (e.g., Score = Fréchet Distance / Step Length or a weighted combination). Explain how the trade-off is calibrated and whether it was tuned on a validation set.

Demonstrate adaptivity empirically:

Include statistics (e.g., histogram or boxplot) showing the distribution of predicted step lengths across different scene types. Do complex scenes trigger longer horizons? This is essential to validate the core claim.

Include ablation studies:

Compare FSN with:

A version using fixed output steps (FSN-Fixed).

A variant without knowledge distillation.

This will isolate the contribution of each component.

Improve baseline comparisons:

Ensure that baseline models (e.g., HiVT, HPNet) are re-implemented fairly and report performance against published leaderboard results.

Report computational efficiency:

Include inference time (ms/sample) for FSN vs. fixed-step models to support claims of efficiency gains.

Release code:

Commit to making the source code and trained models publicly available to enhance reproducibility and impact.

With these revisions, this paper will be a strong contribution to the field of dynamic trajectory prediction.

6. PLOS authors have the option to publish the peer review history of their article (what does this mean?). If published, this will include your full peer review and any attached files.

Reviewer #1: No

Reviewer #2: No

---

## [Author Response · Author response to Decision Letter 1]

9 Sep 2025

Reviewer #1: 1. What is the research subject of this article - vehicle trajectory prediction or pedestrian trajectory prediction? The author needs to be clear.

Response: We appreciate the reviewer for raising this important point. To clarify, The primary research subject of this work is vehicle trajectory prediction in autonomous driving scenarios. This is reflected in the datasets used (Argoverse and INTERACTION), which mainly consist of vehicle trajectories in urban traffic environments. That said, our proposed FlexiSteps Network (FSN) is designed in a general, agent-agnostic manner. The Adaptive Prediction Module (APM) and Dynamic Decoder (DD) operate on latent representations of dynamic agents, and therefore the framework can naturally be extended to other agents such as pedestrians or cyclists.

2. The innovation of this article is not clear. In the contribution section, the author fails to distinguish the differences between the methods presented in this article and those of previous studies, nor does he highlight the problems that this method can solve.

Response: We thank the reviewer for pointing this out. Our work aims to achieve the truth dynamic trajectory prediction, in layman's terms. Because the previous models output the fixed prediction steps whatever is inputed. We have rewrite the expression in p2 of section introduction. Additionally, for the same input sequence, longer prediction lengths lead to greater instability and lower confidence, while shorter prediction lengths lead to shorter reaction times left to the decision system. This leads to the question of a judgmental metric for optimal prediction lengths, we also mentioned “This leads to the question of a judgmental metric for optimal prediction lengths…” in p5 of section introduction. In short, our work gives the model the ability to adaptively judge the output, rather than outputting the same predicted length for all inputs according to a human setting. We have also reorganized the language of the contribution point section.

3. The font in Fig.2 is not uniform. The picture needs to be adjusted. The color of the arrows needs to be darkened. The meanings of some variables in the picture are unclear. Besides, what is the input of the scoring mechanism? It cannot be seen from the picture. Overall, Fig.2 is confusing.

Response: We apologize for the confusion from this figure. We have modified Fig. 2 by unifying the fonts and changing all arrows to black for better clarity and visual consistency. We have enhanced the figure caption and manuscript text to provide clearer explanations of the variables, ensuring that their meanings are now explicitly described. As shown in the upper-right part of Fig. 2, the scoring mechanism takes two inputs: the prediction results from the backbone and the corresponding ground-truth trajectories. To avoid confusion, we have further clarified this in the revised figure and provided a more detailed explanation in Fig. 3 as well as in the Method section (Scoring Mechanism subsection).

4. The full names of APM and DD should be provided for the first time. Additionally, I haven't found the structural details of APM and DD, which prevents me from evaluating the innovativeness of these two modules.

Response: We thank the reviewer for pointing this out. In the revised manuscript, we have made the following changes:

(1) Full names provided: We now explicitly spell out the full names Adaptive Prediction Module (APM) and Dynamic Decoder (DD) when they first appear in the text (see Section Method).

(2) Adaptive Prediction Module (APM): We now detail both its training and inference stages. The APM is a pre-trained two-layer MLP classifier/regressor, which takes the encoded latent features of agents and their contexts as input, and outputs the optimal prediction step. The training procedure is illustrated in Fig.4, and we describe the joint classification and regression losses used to supervise the module.

(3) Dynamic Decoder (DD): We clarify that the DD consists of multiple specialized sub-decoders ${\phi_{D^f}}_{f=5}^F$, each responsible for a specific output step length. During training, the appropriate sub-network is selected according to the APM’s predicted step, and during inference, only the corresponding sub-network is activated. We also emphasize the independent parameter updates of each decoder and our use of KL-divergence regularization to transfer knowledge between different prediction horizons.

These additions ensure that the structural and functional designs of both modules are transparent and evaluable. We believe this will help highlight their novelty and contribution to improving the flexibility of trajectory prediction.

5. The experimental part is insufficient and the comparison methods used are limited. The author claims that the method in this paper improves efficiency, but no analysis of computational cost has been presented.

Response: We sincerely appreciate the reviewer’s insightful comments. In response:

(1) On the sufficiency of experiments:

Following your suggestion, we have conducted additional experiments to strengthen the empirical validation. Specifically, we included results from the FSN-Fixed baseline and further reported computational efficiency metrics to provide a more comprehensive comparison.

(2) On the limited comparison methods:

We acknowledge the concern. However, we would like to note that trajectory prediction for autonomous driving has only recently emerged as an active research area. Compared with fields such as computer vision or natural language processing, the number of closely related baseline models is still relatively limited. The two models we selected for comparison are both recent state-of-the-art works published at CVPR 2024 and NeurIPS 2024, which we believe represent strong and relevant baselines for fair evaluation.

(3) On computational efficiency analysis:

After carefully reviewing the initial submission, we noticed that parts of the manuscript were inadvertently mixed with earlier draft content, which may have caused confusion. We sincerely apologize for this oversight. Initially, we expected that our method might improve computational efficiency; however, as our study progressed, we found that the additional modules introduced inevitably increased computational cost. The final experiments indicate that while the method enhances prediction accuracy, it does not yield significant improvements in efficiency compared with conventional frameworks, as shown in fig3. Accordingly, we have revised the manuscript to clarify this point more accurately. Specifically, in section Abstract, Introduction and Computational Efficiency Analysis of section Main Results.

We hope these revisions address the reviewer’s concerns and improve the clarity and completeness of the experimental section.

6. Many typos exist in this work. Please double-check it.

Response: Thank you for your careful reading of our manuscript. We have performed a thorough, line-by-line proofreading of the entire manuscript (main text, equations, figure/table captions, and headings), followed by an automated spell/grammar pass.

Reviewer #2: The authors propose FlexiSteps Network (FSN), a novel framework for trajectory prediction that dynamically adjusts the number of future time steps predicted based on contextual conditions. This addresses a critical limitation of fixed-horizon models and has strong potential for real-world deployment in autonomous systems.

The motivation is excellent, the use of Fréchet distance for geometric-temporal evaluation is well-justified, and experiments on Argoverse and INTERACTION add credibility. However, several revisions are needed to strengthen the technical contribution and ensure reproducibility:

Clarify the Adaptive Prediction Module (APM):

What is its architecture (e.g., MLP, Transformer)?

Response: Thank you for your careful check. The APM is designed as a two-layer Multilayer Perceptron (MLP), which decodes the latent features produced by the baseline encoder. We intentionally avoided using a more complex Transformer structure to minimize computational overhead during inference. This lightweight design allows APM to be integrated as a plug-and-play module without significantly increasing runtime costs. We also acknowledge that deeper architectures (e.g., Transformer-based) might capture richer associations between contextual features and optimal prediction lengths. This potential extension is explicitly mentioned in the Conclusion as a future research direction.

How is it trained? Are labels for optimal step length generated from ground truth trajectories or heuristics?

Response: As described in the Method section, the training of APM proceeds in three stages:First, we collect multi-modal prediction results from a baseline model trained with different fixed output lengths; These predictions are compared with the corresponding ground truth trajectories, and the scoring mechanism (based on Fréchet distance combined with prediction length) is applied to identify the optimal prediction step length for each agent; These optimal step lengths serve as supervised labels for training the APM. Thus, the labels are derived from ground truth trajectories rather than heuristics.

Is it pre-trained independently or jointly with the decoder?

Response: The APM is pre-trained independently using the procedure described above. It learns to map encoded latent features to optimal prediction steps. The Dynamic Decoder (DD) is then trained separately, but its training relies on the APM’s predicted step lengths during inference. In this way, APM and DD are trained independently, but are integrated sequentially in the overall FSN framework.

Define the scoring mechanism explicitly:

Provide the mathematical formula for the score (e.g., Score = Fréchet Distance / Step Length or a weighted combination). Explain how the trade-off is calibrated and whether it was tuned on a validation set.

Response: We thank the reviewer for the helpful suggestion. In the revised manuscript, we have explicitly provided the mathematical definition of our scoring function in Section Method – Scoring Mechanism. The score is formulated as q_i^f = \frac{d_i^f}{f}, d_i^f = FDK(\mu_i^f, gt_i^f)

where d_i^f denotes the Fréchet Distance Kernel (FDK) between the predicted trajectory \mu_i^f and the ground truth trajectory gt_i^f at prediction horizon f. The optimal prediction length is then obtained as {f_{gt}}_i = \operatorname{argmin}_{f \in \{5,...,F\}} q_i^f.

To address the reviewer’s concern about the trade-off: rather than introducing additional hyperparameters, the division by the step length f naturally calibrates the balance between trajectory similarity and prediction horizon. In this design, longer horizons are only favored when their Fréchet distance is sufficiently low, ensuring principled trade-offs. We also verified the stability of this formulation on the validation set, and it consistently produced reasonable horizon selections without the need for additional tuning.

Demonstrate adaptivity empirically:

Include statistics (e.g., histogram or boxplot) showing the distribution of predicted step lengths across different scene types. Do complex scenes trigger longer horizons? This is essential to validate the core claim.

Response: Thank you for the constructive suggestion. We have added empirical analyses to explicitly demonstrate FSN’s adaptivity:

1.New distribution plots:

We include a new figure titled “Prediction Steps Distribution on Argoverse validation set” (now Fig. 7 in the revised manuscript). It provides: A histogram of predicted step lengths aggregated over the validation set; A 2D heatmap showing the distribution of predicted step lengths across agent-count bins. The x-axis indicates the number of agents in a scene, and the y-axis the predicted horizon; color encodes frequency.

2.Scene-complexity proxy and binning:

We treat the number of agents in a scene as a complexity proxy and report results in bins (e.g., <10, 11–30, 31–50, 51+). This choice is consistent with prior practice and aligns with our data annotations.

3.Findings :

The results show a clear adaptive trend: Simple scenes (few agents) preferentially yield longer horizons (25–30 steps); Moderately complex scenes (11–30 agents) concentrate on medium horizons (15–20 steps) while retaining diversity; Highly complex/congested scenes (≥31 agents, especially 51+) predominantly select shorter horizons (5–15 steps).

Overall, 10–20 steps account for the majority of cases, reflecting a practical balance between stability and foresight.

In our data, more complex scenes do not trigger longer horizons. Instead, they lead to shorter horizons, which is consistent with intuition: higher interaction density and uncertainty warrant nearer-term forecasts with tighter error control, while simpler scenes allow the model to extend its predictive horizon safely.

Include ablation studies:

Compare FSN with:

Response: Thank you for your rigorous consideration. We will respond to each of the following comments.

A version using fixed output steps (FSN-Fixed).

Response: Thank you for this suggestion. In the revised manuscript, we have added FSN-Fixed as an additional baseline in all experiments (see Table 1). This variant disables adaptive step selection by fixing the output length during training and inference, thereby isolating the effect of FSN’s adaptive mechanism. The results show that while FSN-Fixed achieves some improvements over traditional baselines, the full adaptive FSN consistently delivers superior performance, highlighting the importance of dynamic step prediction.

A variant without knowledge distillation.

This will isolate the contribution of each component.

Response: As requested, our ablation study (Table 3 in the revised manuscript) already includes results for both settings: w/ KL denotes training with KL-based knowledge distillation, while w/o KL refers to the variant without knowledge distillation. This provides a direct comparison of the effect of distillation on model performance.

Improve baseline comparisons:

Ensure that baseline models (e.g., HiVT, HPNet) are re-implemented fairly and report performance against published leaderboard results.

Response: Thank you for your rigorous advice. All baseline models (HiVT and HPNet) were re-implemented and evaluated under the same experimental platform, datasets, and hyperparameter settings, as detailed in the Implementation Details section. This ensures fairness and reproducibility in the reported comparisons.

Regarding HPNet, we acknowledge that the reported results differ from the leaderboard numbers. This is because the original HPNet employs a two-stage prediction framework (Stage 1: trajectory coordinate prediction; Stage 2: refinement of predicted coordinates). Due to the large model size and our limited computational resources, we adopted only the first prediction stage, which shares the identical input-processing pipeline with FSN. This design choice was made to (i) ensure fair compatibility with FSN as a plug-and-play module, and (ii) keep training feasible within our resources.

Importantly, when we restored the full HPNet model in a separate experiment, the results matched the published leaderboard, confirming the correctness of our re-implementation. Therefore, the simplified variant used in our main experiments does not compromise the validity of our conclusions, as the goal was to assess FSN’s adaptability rather than re-benchmark HPNet in its entirety.

Moreover, we note that for plug-and-play methods, the key criterion is the relative improvement under a consistent setup, rather than matching absolute leaderboard numbers for every backbone. In fact, for two closely related adaptive-length baselines—LaKD and FlexiLength—their reported results in the original papers also show minor discrepancies from public leaderboard entries, which can arise from differences in preprocessing, map releases, hardware, random seeds, and evaluation tooling. To control for these factors, we re-implement and evaluate all methods within the same codebase, datasets, and hyperparameters, so that the comparative deltas attributable to FSN remain valid and reproducible.

Report computational

---

## [Decision Letter · Decision Letter 1]

22 Sep 2025

Adaptive Output Steps: FlexiSteps Network for Dynamic Trajectory Prediction

PONE-D-25-40072R1

Dear Dr. Niu,

We’re pleased to inform you that your manuscript has been judged scientifically suitable for publication and will be formally accepted for publication once it meets all outstanding technical requirements.

Kind regards,

Jinhao Liang

Academic Editor

PLOS ONE

Additional Editor Comments (optional):

Reviewer #2:

Reviewer #3:

Reviewers' comments:

Reviewer's Responses to Questions

**Comments to the Author**

1. If the authors have adequately addressed your comments raised in a previous round of review and you feel that this manuscript is now acceptable for publication, you may indicate that here to bypass the “Comments to the Author” section, enter your conflict of interest statement in the “Confidential to Editor” section, and submit your "Accept" recommendation.

Reviewer #2: All comments have been addressed

Reviewer #3: All comments have been addressed

2. Is the manuscript technically sound, and do the data support the conclusions?

Reviewer #2: Yes

Reviewer #3: Yes

3. Has the statistical analysis been performed appropriately and rigorously? 

Reviewer #2: N/A

Reviewer #3: Yes

4. Have the authors made all data underlying the findings in their manuscript fully available?

Reviewer #2: Yes

Reviewer #3: No

5. Is the manuscript presented in an intelligible fashion and written in standard English?

Reviewer #2: Yes

Reviewer #3: Yes

6. Review Comments to the Author

Reviewer #2: (No Response)

Reviewer #3: The author has addressed reviewers’ concerns well. I think this paper could be published on this journal.

7. PLOS authors have the option to publish the peer review history of their article (what does this mean?). If published, this will include your full peer review and any attached files.

Reviewer #2: No

Reviewer #3: No

---

## [Editor Report · Acceptance letter]

PONE-D-25-40072R1

PLOS ONE

Dear Dr. Niu,

I'm pleased to inform you that your manuscript has been deemed suitable for publication in PLOS ONE. Congratulations! Your manuscript is now being handed over to our production team.

Kind regards,

on behalf of

Dr. Jinhao Liang

Academic Editor

PLOS ONE